# Early Drought Stress Warning in Plants: Color Pictures of Photosystem II Photochemistry

Michael Moustakas [1,*], Ilektra Sperdouli [1,2] and Julietta Moustaka [1,†]

1   Department of Botany, Aristotle University of Thessaloniki, 54124 Thessaloniki, Greece
2   Institute of Plant Breeding and Genetic Resources, Hellenic Agricultural
    Organisation–Demeter (ELGO-Demeter), Thermi, 57001 Thessaloniki, Greece
*   Correspondence: moustak@bio.auth.gr
†   Current address: Department of Plant and Environmental Sciences, University of Copenhagen,
    Thorvaldsensvej 40, 1871 Frederiksberg C, Denmark.

**Abstract:** Drought, the major limiting factor for plant growth and crop productivity, affecting several physiological and biochemical processes, is expected to increase in duration, intensity, and frequency as a consequence of climate change. Plants have developed several approaches to either avoid or tolerate water deficit. Plants as a response to drought stress (DS), close stomata, reducing carbon dioxide ($CO_2$) entry in the leaf, thus decreasing photosynthesis which results in reduced synthesis of essential organic molecules that sustain the life on earth. The reduced $CO_2$ fixation, decreases electron transport rate (ETR), while the absorbed light energy overdoes what can be used for photochemistry resulting in excess reactive oxygen species (ROS) and oxidative stress. Current imaging techniques allow non-destructive monitoring of changes in the physiological state of plants under DS. Thermographic visualization, near-infrared imaging, and chlorophyll *a* fluorescence imaging are the most common verified imaging techniques for detecting stress-related changes in the display of light emission from plant leaves. Chlorophyll *a* fluorescence analysis, by use of the pulse amplitude modulation (PAM) method, can principally calculate the amount of absorbed light energy that is directed for photochemistry in photosystem II (PSII) ($\Phi_{PSII}$), dissipated as heat ($\Phi_{NPQ}$), or dissipated by the non-radiative fluorescence processes ($\Phi_{NO}$). The method of chlorophyll *a* fluorescence imaging analysis by providing colour pictures of the whole leaf PSII photochemistry, can successfully identify the early drought stress warning signals. Its implementation allowed visualization of the leaf spatial photosynthetic heterogeneity and discrimination between mild drought stress (MiDS), moderate drought stress (MoDS), and severe drought stress (SDS). The fraction of open reaction centers of PSII (q*p*) is suggested as the most sensitive and suitable indicator of an early drought stress warning and also for selecting drought tolerant cultivars.

**Keywords:** chlorophyll fluorescence imaging; plant phenotyping; photosynthetic heterogeneity; climate change; reactive oxygen species; mild drought stress; moderate drought stress; severe drought stress; acclimation; abiotic stress

## 1. Introduction

Plant growth and development experience a non-stop exposure to biotic and abiotic stresses conditions. Photosynthesis is the device of crop productivity, but besides it is a complicated process that is highly responsive to biotic and abiotic stresses with a complex association to plant growth [1]. In this review article we present the impact of water deficiency on plants and the drought avoidance and drought tolerance mechanisms of plants. We discuss on the drought stress effects on photosynthesis with special emphasis on the light reactions of photosynthesis and the production of reactive oxygen species in photosystems I (PSI) and II (PSII). We present a theoretical background of the method of chlorophyll fluorescence analysis that is used for the evaluation of photosynthetic function

under abiotic and biotic stress conditions and we mention some other imaging techniques that allow a non-destructive monitoring of changes in the physiological state of plants under drought stress. As an application example of the method of chlorophyll fluorescence imaging analysis in drought stress phenotyping, we selected the model plant *Arabidopsis thaliana* as the most suitable plant species, and we provide colour pictures of the whole leaf PSII photochemistry that can be used successfully as early drought stress warning signals.

Drought is one of the major limiting factors for plant growth and crop productivity because it affects several physiological and biochemical processes [2–5] and is expected to increase in duration, intensity, and frequency as a consequence of climate change [6–10], reaching to alarm level by now days [11–13]. Drought stress (DS) is the main issue amongst all environmental situations associated with the forecast effects of climate change that will detrimentally influence global crop production [14,15]. Water deficit impairs plant's cell division, elongation and differentiation, osmotic adjustment, causing loss of turgor, and harms photosynthetic rates and growth, disturbing energy balance, and eventually decreases plant productivity [16–18].

As a response to DS, plants close stomata to reduce water loss (transpiration) in order to prevent dehydration, but this results in limiting $CO_2$ to penetrate the leaf, thus affecting detrimentally photosynthesis [2,3]. Thus, stomatal aperture must be strictly regulated to play a dual role, preventing dehydration while sustaining photosynthesis [2,3]. Photosynthesis is fundamental to plant growth, functioning and fitness, however the plant's capability to acquire and retain highest photosynthetic capacity greatly relies upon the environmental conditions [19,20]. Photosynthesis of food crops under DS has been considered to be a real challenge for plant scientists and crop breeders in order to fulfill the huge demand for food in the world [21,22]. Photosynthetic capacity and plant productivity are frequently superior in environmental conditions with higher water availability since $H_2O$ accomplishes an essential role in photosynthesis [3,20]. The oxidation of $H_2O$ molecules in Photosystem II (PSII), that uses the light energy, provides protons ($H^+$), and electrons ($e-$) that result in the formation of ATP and reducing power (reduced ferredoxin and NADPH) (Figure 1), for the synthesis of essential organic molecules that sustain the life on Earth [23–25]. PSII supramolecular complex consists of a water-splitting system (oxygen evolving complex, OEC), a light-harvesting chlorophyll protein complex (LHCII) and a reaction center (RC).

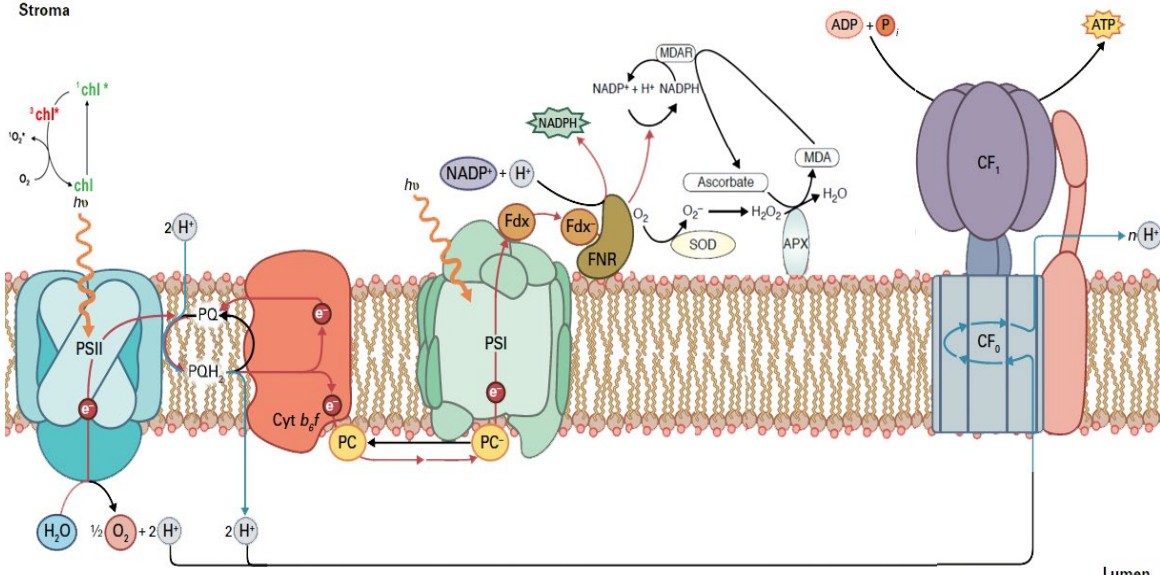

**Figure 1.** Light energy capture and utilization in the thylakoids of chloroplasts. The electron transport chain from photosystem II (PSII) to photosystem I (PSI) and finally to ferredoxin (Fdx) to form NADPH, is depicted. Detail explanation in the text (Adopted from [26]).

Light-harvesting complex of PSII (LHCII) absorbs light energy and transfers it to the RC of PSII. When there is an excess light energy, creation of singlet oxygen ($^1O_2$) via the triplet state of chlorophyll ($^3chl*$) occurs [27] (Figure 1). At PSII the oxidation of water, at the water-splitting complex, results to $O_2$, $H^+$, and e− (Figure 1). The e− are transferred from $H_2O$ to $NADP^+$, while accompanying this e− transfer, a proton gradient is established across the membrane that is utilized for the synthesis of ATP by the ATP synthase [28]. At the RC of PSII the excitation of specially bound chlorophyll molecules results in transfer of an e− from $H_2O$ oxidation to quinone A ($Q_A$). The fully reduced quinol molecule ($PQH_2$) picks up two $H^+$ from the stroma and is oxidized to a quinone (PQ) and while the e− are transferred through cytochrome $b_6f$, to plastocyanin (PC) and to PSI, $H^+$ are transferred from the stroma to the chloroplast lumen [28] (Figure 1). The structures of the soluble proteins ferredoxin (Fdx) and ferredoxin-$NADP^+$ reductase (FNR), on the stromal side, that transfer the e− to $NADP^+$ to form NADPH, are also shown in Figure 1. As a result of DS, stomatal closure, limits $CO_2$ entry to the leaf, and since NADPH is not used in Calvin–Benson–Bassham cycle, $NADP^+$ is not available. Under such circumstances, e− are transferred to molecular oxygen ($O_2$) forming superoxide anions ($O_2^{\bullet-}$) that are converted by the superoxide dismutase (SOD) to hydrogen peroxide ($H_2O_2$) that is reduced by ascorbate peroxidase (APX) to $H_2O$ and $O_2$ [29,30] (Figure 1). Ascorbate peroxidase uses electrons from ascorbate (AsA) that is oxidized, but through monodehydroascorbate reductase (MDAR), AsA is reduced from NADPH, contributing to $NADP^+$ availability [29] (Figure 1).

## 2. Drought Stress Impact on Plants

Plant metabolism is accomplished with water involvement, and adequate water is an important circumstance for growth and development [31]. Drought stress that negatively impacts plant growth is the major constraint to crop production and a growing concern for crop yields as a result of global climate changes that involve increased DS periods [32,33]. Plants are subjected to DS conditions when either the water supply to the roots is restricted or the water loss through transpiration is extreme [34]. Thus, the imbalance from a reduced water uptake with an excessive water loss, and the oxidative damage stimulated by the increased ROS generation, that are induced by water deficiency, result in noteworthy changes in plant growth, biomass production, photosynthesis, and enzymatic activities [34–40].

Drought stress impairs osmotic adjustment of plants and harms photosynthesis and growth [17,41,42], resulting in reduced crop yields that affect food security worldwide [7,43,44], with up to 21 and 40% yield reductions in wheat (*Triticum aestivum* L.) and maize (*Zea mays* L.), respectively [45]. Stomatal closure, as a response to water deficit, reduces excess water loss (transpiration) to prevent desiccation, but also limits $CO_2$ to penetrate the leaf, thus harmfully affecting photosynthesis [46]. Due to this compromise, stomatal aperture must be strictly coordinated [33,47].

The unfavorable conditions of climate change are contributing to development of extended water deficit areas and consequently on the plant growth and crop productivity. Although a positive impact of elevated $CO_2$ on crop yield by increasing photosynthesis is suggested by some researchers, it is debated by some others showing that increased $CO_2$ does not counteract the effect of severe drought on photosynthesis and yield [46,48].

## 3. Plant Tolerance to Drought Stress

Plants have developed several energetic approaches at the morphological, physiological and biochemical levels, permitting them to avoid and/or tolerate water deficit [9,42,46,49,50]. Avoidance mechanisms are mainly morphological and physiological adjustments that provide an escape to the water deficit, e.g., by increased root system, increased leaf thickness, decreased leaf area, reduced stomatal number and conductance, and leaf rolling or folding to minimize evapotranspiration [46,51,52]. Drought tolerance traits are correlated with maintenance of the plant water status throughout osmotic adjustment by the accumulation

of osmoprotective substances, for example, proline, glycine betaine and sugars, that help the plants to preserve their water status [41,42,53–56], and to acclimate to water deficit [9,42,57].

Drought stress can develop in a wide range, from mild drought stress (MiDS), to moderate drought stress (MoDS), and to severe drought stress (SDS), thru which plants experience dehydration and wilting, which ultimately leads to death [9,58]. Numerous studies have focused on SDS, although, MoDS is developing more frequently in actual circumstances [58]. Current studies have confirmed that plants employ diverse approaches to manage MiDS, compared to MoDS or to SDS [9,58–63]. For instance, in Arabidopsis young leaves photosynthetic efficiency is recovering under MoDS, but not in MiDS [38,64], while plants that were defined to be tolerant to SDS do not withstand MiDS [61].

Plant responses to DS commonly fluctuates from species to species depending on plant growth and developmental stage and also from other environmental factors [32,36,38,64].

## 4. Photosynthetic Function under Drought Stress

Drought stress significantly decreases photosynthetic activity and disrupts plant productivity [65]. The decreased photosynthetic activity is linked to both stomatal and non-stomatal effects, which are not totally understood [55,65–68]. Drought stress decreases photosynthesis by reducing carbon dioxide availability through increased resistance to carbon dioxide flow from stomata, disrupts either biochemical or/and photochemical activity and increases leaf membrane lipid peroxidation [9,37,38,69–72]. The diminished $CO_2$ fixation under DS results in decreased electron transport rate (ETR) [73,74], while the absorbed light energy that overdoes what it can be used for photochemistry results in surplus reactive oxygen species (ROS) accumulation, that can harm the chloroplast, and particularly damage photosystem II (PSII) [9,24,75–77]. However, damage of PSII can be prevented by dissipation of excess light energy as heat, a process termed non-photochemical quenching (NPQ), and typically estimated by chlorophyll *a* fluorescence analysis [9,78–80]. The NPQ mechanism is considered to be the principal photoprotective mechanism and is more efficient in the combined existence of the PsbS (PSII subunit S) protein, and zeaxanthin of the xanthophyll cycle [81,82]. NPQ by dissipating harmless the excess absorbed light energy under DS, decreases ETR to prevent ROS formation, thus it can regulate to a level ROS formation [83–85]. Our understanding of the NPQ process has been advanced by use of the pulse amplitude modulation (PAM) fluorescence analysis that it is quantifying NPQ photoprotective potential in addition to the classical chlorophyll fluorescence induction analysis [86].

## 5. Reactive Oxygen Species Generation under Drought Stress

ROS, such as superoxide anion radical ($O_2^{\bullet-}$), hydrogen peroxide ($H_2O_2$), and singlet oxygen ($^1O_2$), are continuously produced at basal levels, mainly in the light reactions of photosynthesis, but are kept in a homeostasis by the antioxidative enzymatic and non-enzymatic systems [17,30,79,87]. Drought stress breakdown the balance between the creation and elimination of ROS in plants [9]. Thus, during water deficit periods ROS creation rises extremely [88], and this triggers oxidative stress causing membrane injuries, protein degradation and enzyme inactivation that damage the cellular components [40,89,90]. To prevent oxidative damage, beside the NPQ mechanism that is considered as the principal photoprotective mechanism, plants have an effective antioxidant defence system, with both enzymatic and nonenzymatic systems [40,91,92]. Efficient enzymatic antioxidants, such as SOD, APX, MDAR, glutathione reductase (GR), glutathione peroxidase (GPX), guaiacol peroxidase (GOPX), and catalase (CAT), and the non-enzymatic metabolites, such as AsA, glutathione (GSH), a-tocopherol, carotenoids, phenolic compounds, flavonoids, and proline [93–95], play critical roles in removing the water deficiency-induced excessive ROS [96]. The ascorbate-glutathione (AsA-GSH) cycle is a crucial component of the enzymatic antioxidant defence system in plants [97,98].

The most reactive of all ROS is the hydroxyl radical ($OH^\bullet$) that reacts with almost all molecules but it is the shortest lived. The e− leakage to $O_2$ at PSI results in $O_2^{\bullet-}$ which

is rapidly converted by SOD to $H_2O_2$ that is longer lived than $O_2{}^\bullet-$, which is shorter lived than $^1O_2$ but longer lived than $OH^\bullet$. Hydrogen peroxide is the most stable and least reactive ROS with the longest lifetime, and is being able to easily diffuse through the membranes [29,30,99–101].

ROS production in the process of light absorption and energy use in photosynthesis confers an important biological function (plant growth and development, redox signaling) besides generating oxidative stress [99–102]. The role of antioxidants (enzymic and non-enzymic) in the photosynthetic apparatus is not to totally remove ROS, but rather to accomplish an appropriate equilibrium among production and scavenging so as to pair the process of photosynthesis, permitting an efficient spreading of signals to the nucleus [103–105].

ROS function is not only to monitor electron transport and, consequently, prevent over-reduction or over-oxidation, but also generate redox regulatory networks that allow plants to sense and react to environmental stress conditions [88,104,106,107]. ROS trigger plant's protection mechanisms in order to manage oxidative stress, and are now considered important signaling molecules for the regulation of plethora physiological functions and the acclimation response [65,85,87,88,98,107–111].

Under DS, an excess accumulation of ROS can damage the chloroplast, with PSII being especially exposed to damage [9,24,65,77]. Although ROS were primarily considered to be toxic by-products it is now recognized that a basal level of ROS is fundamental to sustain life [99,112,113]. A basal level of ROS is desirable for optimal plant growth, while a little amplified level of ROS is beneficial for triggering stress defense responses, but a high level of ROS out of the limits is considered harmful to plants [25,103]. Nowadays, the consequences of global climate change request a better understanding of the relationship between PSII photochemistry and ROS role as a molecule for photoprotection [101]. The illumination of this interaction could assist to enhance agricultural sustainability under a global climate change [101].

## 6. Plant Phenotyping for Drought Stress Tolerance

Climate change quickly turns-off into a climate crisis with huge worries for agricultural production. Since water deficit is one of the key hazards for the future of agriculture and the total worldwide population, assessing and investigating the ability of crops to grow with restricted water is therefore essential [65,114]. In recent years, much effort has been made to study plant responses to drought in order to address the present and future risks associated with climate change [65,115,116]. The availability of diverse imaging techniques has allowed real-time imaging analysis of physiological changes in plants under DS for high-throughput screening [117–119]. The development of modern analytical techniques has created huge inputs to high-throughput plant phenotyping providing various information related to plant status [120]. Selecting the proper imaging sensors is fundamental in designing phenotyping facilities, which depend on the special experimental objects [121]. These techniques allow the pre-symptomatic monitoring of plant stress, a long time before any visible symptoms developed, enabling for high-throughput screening [117]. By the time that visible symptoms of stress are displayed, the plant could have been already adversely affected [117]. Current imaging techniques allow non-destructive monitoring of changes in the physiological state of plants under DS [37,120,122]. Thermal imaging (also called far-infrared thermal imaging) that determines energy loss from stomatal aperture by the leaf temperature, hyperspectral imaging (visible and near-infrared) that can provide spatial information simultaneously, and chlorophyll *a* fluorescence imaging are the most common verified imaging techniques for detecting stress-related changes from plant leaves [117,121–128]. Crop monitoring using imaging techniques would allow us to relieve stress at an early stage, avoiding permanent damage and thus considerably decreasing yield losses [117,120,129].

Chlorophyll *a* fluorescence imaging uses blue and red fluorescence to detect the emission that results from absorbed light energy which is not dissipated as heat, or it is not used

for the photosynthetic reactions in photochemistry [122]. The chlorophyll fluorescence parameters that are measured can be decoded in terms of photosynthetic activity to acquire knowledge about photosynthetic function and particularly of PSII [130,131]. The data of chlorophyll *a* fluorescence measurements have been considerably used to probe the function of the photosynthetic apparatus and for screening different crops for plant tolerance to numerous stresses [122,132–144]. Chlorophyll fluorescence imaging instruments offer the option to evaluate photosynthetic function at the whole leaf surface and identify leaf spatial heterogeneity [122,145,146]. Photosynthetic performance is extremely heterogeneous at the leaf surface, especially under stress conditions [145,147–153]. Chlorophyll fluorescence analysis is a quick, easy, non-invasive, cheap, and highly sensitive method that can determine photosynthetic function accurately and sense the impact of different stresses on plants [122,151,154].

## 7. Theoretical Aspects of Chlorophyll Fluorescence Analysis

Chlorophyll *a* fluorescence analysis, with the use of pulse amplitude modulation (PAM) method can principally calculate the amount of absorbed light energy that is directed to PSII for photochemistry, dissipated as heat through the non-photochemical quenching (NPQ) mechanism, or it is dissipated by the less well characterized non-radiative fluorescence processes, that are marked as $\Phi_{PSII}$, $\Phi_{NPQ}$, and $\Phi_{NO}$, respectively, with the sum of them to be equal to one [122,155,156].

Chlorophyll fluorescence quenching analysis using the PAM method is based on the principle that the leaf has to be in a dark-adapted state before the measurements, so as the first stable acceptor of PSII, $Q_A$, is fully oxidized. This can be achieved by dark incubation for several minutes [154]. In the dark-adapted state a "measuring light" is switched on that is high enough to produce the minimal level of chlorophyll fluorescence, termed F*o* (Figure 2). A short-lived saturating pulse of light results in the formation of the maximum yield of fluorescence, F*m*. By subtracting F*o* from F*m* the variable fluorescence, F*v* results. The ratio F*v*/F*m* is indicator of the maximum quantum yield of PSII photochemistry. The application of saturating pulses under actinic light illumination closes all the reaction centers and provides the maximum fluorescence in the light-adapted state, that is termed F*m'*. The steady-state level of fluorescence in the light is the F*s* and is calculated before switching off the actinic light. F*o'* is measured directly after switching off the actinic light. By subtracting F*o'* from F*m'* the variable fluorescence, F*v'* results. The ratio F*v'*/F*m'* is indicative of the efficiency of excitation energy capture by open PSII reaction centers.

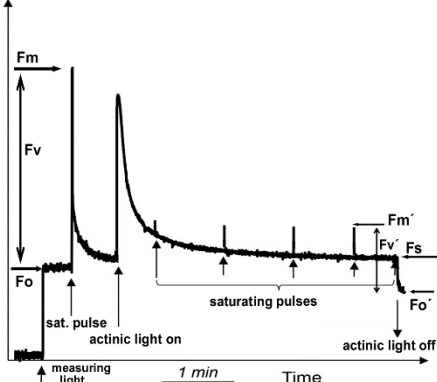

**Figure 2.** A typical modulated fluorescence trace using dark-adapted leaf material showing how F*o*, F*m*, F*o'*, F*m'* and F*s*, are formed to measure chlorophyll fluorescence parameters. In the dark-adapted state a "measuring light" of low light intensity is switched on to induce electron transport through PSII and to elicit the minimal level of chlorophyll fluorescence, termed F*o*. A brief saturating pulse of light results in the formation of the maximum yield of fluorescence, F*m*. The difference between F*m* and F*o* is the variable fluorescence, F*m*. The application of saturating pulses under actinic light illumination closes all the reaction centers and provides the maximum fluorescence in the light-adapted

state, termed F*m*′. The steady-state level of fluorescence in the light is termed, F*s* and is measured immediately before switching off the actinic light. F*o*′ is measured immediately after switching off the actinic light. The difference between F*m*′ and F*o*′ is the variable fluorescence, F*v*′ (Adopted from [26]).

Since photosynthetic performance is not homogeneous at the leaf surface, especially under DS conditions, it renders conventional chlorophyll fluorescence analysis non-representative of the physiological status of the whole leaf [145,149,150]. The manufactured new chlorophyll fluorescence imaging instruments that are capable of analysing the spatial photosynthetic heterogeneity at the whole leaf surface offer new potentials that cannot be obtained by conventional chlorophyll fluorescence analysis [126,127,138,145]. A list of chlorophyll *a* fluorescence parameters used in studies of photosystem II photochemistry, and in this review, together with their definitions, are given in Table 1.

**Table 1.** Chlorophyll fluorescence parameters used in studies of photosystem II photochemistry with their definitions.

| Parameter | Definition | Calculation |
|---|---|---|
| F*v*/F*m* | Maximum efficiency of PSII photochemistry | (F*m* − F*o*)/F*m* |
| $\Phi_{PSII}$ | Effective quantum yield of PSII photochemistry | (F*m*′ − F*s*)/F*m*′ |
| $\Phi_{NPQ}$ | Quantum yield of regulated non-photochemical energy loss in PSII | F*s*/F*m*′ − F*s*/F*m* |
| $\Phi_{NO}$ | Quantum yield of nonregulated energy loss in PSII | F*s*/F*m* |
| F*v*′/F*m*′ | Efficiency of open PSII centers | (F*m*′ − F*o*′)/F*m*′ |
| F*v*/F*o* | Efficiency of the oxygen evolving complex (OEC) on the donor side of PSII | (F*m* − F*o*)/F*o* |
| ETR | Electron transport rate | $\Phi_{PSII} \times$ PAR $\times$ c $\times$ abs, where PAR is the photosynthetically active radiation, c is 0.5, and abs is the total light absorption of the leaf taken as 0.84 |
| q*p* | Photochemical quenching, representing the fraction of PSII reaction centers in open state (puddle model) | (F*m*′ − F*s*)/(F*m*′ − F*o*′) |
| NPQ | Non-photochemical quenching reflecting the dissipation of excitation energy as heat | (F*m* − F*m*′)/F*m*′ |
| EXC | Excess excitation energy | (F*v*/F*m* − $\Phi_{PSII}$)/F*v*/F*m* |
| 1−q*L* | The fraction of PSII reaction centers in closed state (based on a "lake" model for the photosynthetic unit) | q*p* × F*o*′/F*s* |

## 8. Chlorophyll Fluorescence Analysis for Drought Stress Tolerance

Chloroplasts throughout the procedure of photosynthesis play an essential role as redox sensors of DS conditions and stimulate acclimatory or stress defense responses [38,157–160]. The redox state of $Q_A$ has a critical influence on plant growth, development, and defence [161], and is considered as a sensor of the energy imbalance under any stress conditions [162,163]. If the excess energy can not be dissipated under stress conditions, over-reduction of the photosynthetic electron transport chain (ETC) occurs [156]. Over-reduction of the ETC can severely damage the chloroplast and the cell [164]. Excess excitation energy and consequently an imbalance between energy supply and demand outcomes in increased ROS production [84,105,110], that causes damage to proteins, lipids, and nucleic acids [156,165,166]. Photoinhibition is a product of this damage, and PSII is the primary photoinhibition target [167]. Photoinhibition reduces the number of active PSII centers and is widespread across species, light situations and habitats [168–170]. It can be estimated by chlorophyll *a* fluorescence analysis, based on the ratio F*v*/F*m*, the maximum efficiency of PSII photochemistry [167,169].

The redox state of $Q_A$, as estimated by the parameter $1 - qL$, is representing the fraction of PSII reaction centers in closed state (based on a "lake" model for the photosynthetic unit) [155]. Changes in the redox state of $Q_A$ are considered to act as a signal to the stomatal guard cells [65,163]. A more oxidized $Q_A$ pool under DS conditions matches to the lowest stomatal opening and it is linearly correlated to stomatal conductance [65,171], that is a measure of stomatal closure and is commonly used as a water stress index [172,173]. It is

now broadly accepted that the redox signals are important regulators of plant metabolism and also death [174,175].

Using the model plant *Arabidopsis thaliana*, that is considered as the most suitable for the application of the method of chlorophyll fluorescence imaging analysis [37], the early DS responses of photosynthesis were assessed (Figures 3 and 4). Between the chlorophyll fluorescence parameters that have been used for evaluation, monitoring, and selection of drought-tolerant plants, the maximum efficiency of PSII photochemistry ($Fv/Fm$) was the one that was used most [72]. The photosynthetic efficiency of *Arabidopsis thaliana* plants, whose watering stopped (i) twenty-four hours before sampling, characterized as being at the onset of drought stress (OnDS); (ii) six days before sampling, characterized as at mild drought stress (MiDS); and (iii) ten days before sampling, characterized as at moderate drought stress (MoDS) [37], was evaluated by chlorophyll fluorescence imaging analysis. The maximum efficiency of PSII photochemistry ($Fv/Fm$) decreased as soon as 24 h after the onset of drought stress (OnDS) (Figure 3b), while under MiDS decreased further (Figure 3c). However, under further water deficit treatment (ten days, MoDS), the maximum efficiency of PSII photochemistry recovered (Figure 3d). Further DS treatment, characterized as severe drought stress (SDS), resulted in a significant diminished PSII photochemistry [37]. Most authors did not find significant decreases in $Fv/Fm$ under MoDS [3,176,177], indicating that ETR is unaltered under MoDS [178]. The reduction status of the plastoquinone pool ($qp$) in *Arabidopsis thaliana* decreased as soon as 24 h after the onset of drought stress (OnDS) (Figure 4b) by 18%, while under MiDS by 66% compared to well-watered Arabidopsis plants (Figure 4c). However, under MoDS $qp$ was 34% lower than well-watered Arabidopsis (Figure 4d). Photosystem II reaction centers are supposed to be open ($qp = 1$) or closed ($qp = 0$) depending upon whether they are ready to accept light energy from antennas to excite an electron (open), or unable to accept light energy (closed). Based on the abovementioned data the fraction of open reaction centers of PSII ($qp$), is a more sensitive parameter to probe DS effects compared to $Fv/Fm$. Recent studies proposed that the fraction of open reaction centers of photosystem II (PSII) ($qp$), or in other words the reduction status of the plastoquinone pool, is more sensitive than the $Fv/Fm$ that is traditionally used, and thus $qp$ is a more proper indicator to probe the effects of biotic or abiotic stresses on leaf photosynthesis [9,105,179], and to select drought tolerant cultivars under deficit irrigation [65].

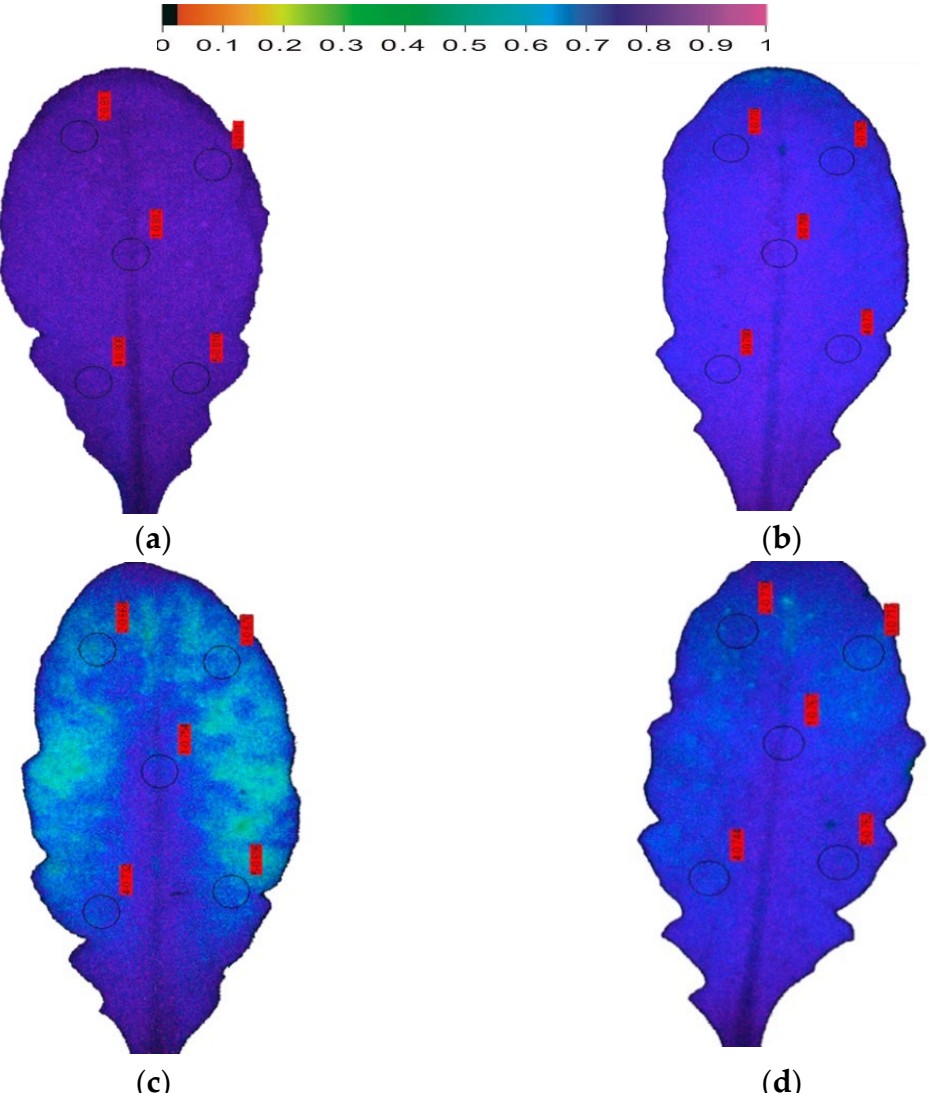

**Figure 3.** Color-coded leaf images of the maximum efficiency of *Arabidopsis thaliana* PSII photochemistry ($Fv/Fm$). $Fv/Fm$ decreased from the control value of 0.811 (**a**), with the onset of drought stress (OnDS) to 0.775 (**b**), and decreased further to 0.714 under mild drought stress (MiDS) (**c**); while under further water deficit treatment, $Fv/Fm$ recovered to 0.743 at moderate drought stress (MoDS) (**d**).

The resulting increase in leaf spatial heterogeneity from well-watered (control plants) to MiDS is reflected in the decrease of the fraction of open reaction centers ($qp$) integrated over the leaf (Figure 4). The increase of the fraction of open reaction centers ($qp$) under MoDS retained the leaf spatial heterogeneity. After exposure to DS, $qp$ values decreased differentially in different parts of the leaf. Under water deficit treatments $qp$ values were higher in the proximal (leaf base) compared to the distal (leaf tip) part (Figure 4). Under all DS treatments, $Fv/Fm$ decreased less in the leaf base than in the leaf tip (Figure 3). Decline of the $Fv/Fm$ ratio has been related to a decline in the ability of PSII to reduce the primary electron acceptor, $Q_A$ [180].

The spatiotemporal heterogeneity observed in *A. thaliana* leaves under DS implies that pigment concentration and composition, water potential and stomatal function unquestionably fluctuate in different cell regions of the leaf, contributing to spatial differentiations in photochemical activity [181,182]. This spatial photosynthetic heterogeneity under DS may reflect diverse zones of leaf anatomy and mesophyll development [37]. Blade maturation of Arabidopsis leaves appears from the tip to the base of the leaf [37], the latter representing younger cells in leaf anatomy [183], with not fully developed chloroplasts [184].

While *A. thaliana* does not seem to activate any tolerance mechanism under MiDS, it appears not to suffer under MoDS [37]. This tolerance mechanism under MoDS was suggested to be activated by an antioxidant defense mechanism that activated ROS scavenging [9]. An early drought warning system is much more than a forecast for decision making in response to a changing climate [185,186].

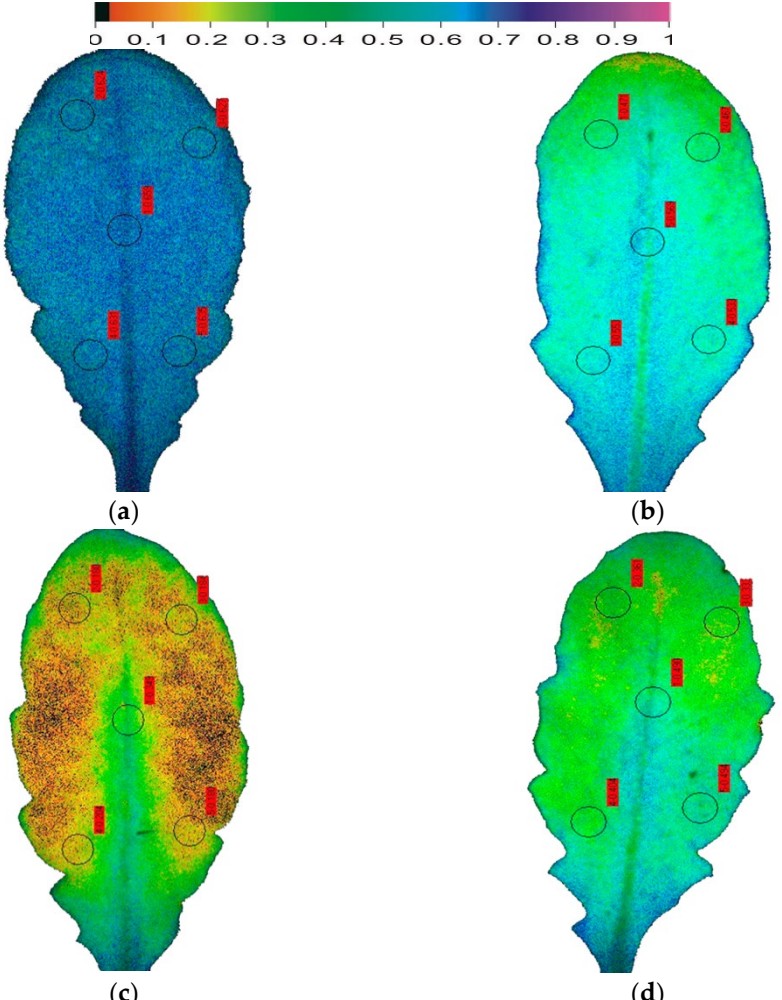

**Figure 4.** Color-coded images of the fraction of open PSII reaction centres ($qp = [Fm' - Fs]/[Fm' - Fo']$), representing the redox state of $Q_A$, in *Arabidopsis thaliana* leaves. The fraction of open PSII reaction centres ($qp$) decreased from the control value of 0.635 (**a**), with the onset of drought stress (OnDS, 24 h DS) to 0.519 (**b**), under mild drought stress (MiDS) decreased further to 0.218 (**c**); while under moderate drought stress (MoDS), $qp$ recovered to 0.416 (**d**).

## 9. Conclusions

The method of chlorophyll fluorescence analysis has been widely used to monitor biotic and abiotic stress effects on plants by using leaf photosynthesis attributes [130,135,137,154,187]. Recently, besides the above-ground parts of crops, a phenotyping method focused on the rooting system and its activity was proposed [188]. The color pictures of PSII photochemistry that can be obtained by chlorophyll fluorescence imaging analysis [37,122] are suitable to characterize and differentiate plant tolerance to DS by evaluating their photosynthetic efficiency and produce also useful information that can be used effectively to phenotype plants under water deficit, with the aim to identify the optimum irrigation conditions. The method of chlorophyll *a* fluorescence imaging analysis by providing colour pictures of the whole leaf PSII photochemistry, can successfully identify the early DS warning signals allowing the pre-symptomatic monitoring of DS in a non-destructive way. It is an easy,

quick, cheap, non-invasive, and highly sensitive method [122,187]. Its implementation allowed visualization of the leaf spatial photosynthetic heterogeneity and discrimination between mild drought stress (MiDS), moderate drought stress (MoDS), and severe drought stress (SDS) [37]. The reduction status of the plastoquinone pool or in other words the fraction of open reaction centers of PSII ($qp$) has been shown to be the most sensitive and suitable indicator to probe photosynthetic function and determine the impact of biotic and abiotic stresses on plants [37,105,189], and also to select drought tolerant cultivars under deficit irrigation [65].

**Author Contributions:** Conceptualization, M.M.; software, J.M.; validation, M.M., I.S. and J.M.; formal analysis, M.M., I.S. and J.M.; data curation, M.M., I.S. and J.M.; writing—original draft preparation, M.M.; writing—review and editing, M.M., I.S. and J.M.; visualization, M.M.; supervision, M.M.; project administration, M.M. All authors have read and agreed to the published version of the manuscript.

**Funding:** This research received no external funding.

**Data Availability Statement:** The data presented in this study are available in this article.

**Conflicts of Interest:** The authors declare no conflict of interest.

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
