# Peer review of "Early Drought Stress Warning in Plants: Color Pictures of Photosystem II Photochemistry"

_climate, doi:10.3390/cli10110179_

Round 1

Reviewer 1 Report

The review entitled “Early Drought Stress Warning in Plants: Color Pictures of Photosystem II Photochemistry” described about role of PSII in the early drought stress and this potential to be explored as warning signals.

The Review is good and covers the unique topic. Plants behave differentially to each other under stress and also a set of parameters which explain plant- attributes cannot be apply for others. It is always better that such parameters may generalized and explained with examples and also deviation may discuss.   I feel that some key physiological parameters may be provided as early indicator of the drought for a particular family of plants.

Author Response

The review entitled “Early Drought Stress Warning in Plants: Color Pictures of Photosystem II Photochemistry” described about role of PSII in the early drought stress and this potential to be explored as warning signals.

The Review is good and covers the unique topic. Plants behave differentially to each other under stress and also a set of parameters which explain plant- attributes cannot be apply for others. It is always better that such parameters may generalized and explained with examples and also deviation may discuss. I feel that some key physiological parameters may be provided as early indicator of the drought for a particular family of plants.

Thank you for your comment that helped us to improve our manuscript. We included in the Abstract (lines 29-31) and in Conclusion (lines 400-404) our suggestion of a key physiological parameter that can be used as an early indicator of drought

Reviewer 2 Report

Drought has been a critical problem of ecosystem and crop production under climate change. The estimation of the drought to generate accurate, fast, and effective information for farmers is also important to conduct suitable management solutions such as watering. The topic is attractive to show the early drought stress warning in plants through the color picture. The paper is overall well-written and it has the potential to be accepted, but after some revisions.

The abstract has well described the advantages of the technique, but two much. I suggest authors should add some description on the technique mechanism and the demonstration of the application.

Introduction, it is good to see authors trying to describe the mechanism behind the color analysis. However, what is the research gap?

A method section to show the work of this paper is needed. Authors have not followed a scientific paper pattern.

Is section 8 a case study? How to show the accuracy of your model? This should be formally presented. Methods, study period and assessment indicators should be presented.

The following references are useful.

3D characterization of crop water use and the rooting system in field agronomic research. Computers and Electronics in Agriculture, 202, 107409.

Soil exchangeable cations estimation using Vis-NIR spectroscopy in different depths: Effects of multiple calibration models and spiking. Computers and Electronics in Agriculture, 182, 105990.

Author Response

Drought has been a critical problem of ecosystem and crop production under climate change. The estimation of the drought to generate accurate, fast, and effective information for farmers is also important to conduct suitable management solutions such as watering. The topic is attractive to show the early drought stress warning in plants through the color picture. The paper is overall well-written and it has the potential to be accepted, but after some revisions.

The abstract has well described the advantages of the technique, but two much. I suggest authors should add some description on the technique mechanism and the demonstration of the application.

Yes, the Abstract may be too much. However, we included (lines 29-31) an application of the method as you suggested.

Introduction, it is good to see authors trying to describe the mechanism behind the color analysis. However, what is the research gap?

We tried to describe only the mechanism and the application of chlorophyll fluorescence analysis without dealing with technical matters that can be found in operation manuals of the instrument. However, following your comment, we inserted in Introduction the objectives of our review (lines 40-52).

A method section to show the work of this paper is needed. Authors have not followed a scientific paper pattern.

Since our manuscript is a review manuscript, we did not include a method section that is found in research articles.  

Is section 8 a case study? How to show the accuracy of your model? This should be formally presented. Methods, study period and assessment indicators should be presented.

Since the method of chlorophyll fluorescence imaging analysis has the capability to distinguish drought symptoms as early as 24 h of water deficit it seems it can be regarded as accurate. In Conclusion (lines 400-404) we suggest of a key physiological parameter that can be used as an early drought assessment indicator.

The following references are useful.

3D characterization of crop water use and the rooting system in field agronomic research. Computers and Electronics in Agriculture, 202, 107409.

Soil exchangeable cations estimation using Vis-NIR spectroscopy in different depths: Effects of multiple calibration models and spiking. Computers and Electronics in Agriculture, 182, 105990.

We included the second reference that you suggested in our revised manuscript.

Author Response

Comment 1 page 1 (lines 14-16).

The sentence was improved.

Comment 2 page 1 (line 40).

We change it (line 53) to : “Drought is one of the major limiting factors”

Comment 1 page 2 (lines 49-52).

We wrote the sentence (lines 62-65): “As a response to DS, plants close stomata to reduce water loss (transpiration) in order to prevent dehydration, but this results in limiting CO2 to penetrate the leaf, thus affecting detrimentally photosynthesis [2,3]. Thus, stomatal aperture must be strictly regulated to play a dual role, preventing dehydration while sustaining photosynthesis [2,3].”

Comment 2 page 2 (line 76) that continues with Comment 1 page 3 (line 77).

We explained it in lines 62-65.

Comment 2 page 3 (lines 81-83)

Since our manuscript is a review manuscript focused on the use of chlorophyll fluorescence imaging analysis as an early drought stress warning signal, we did not formulate a hypothesis and we did not follow the common structure of introduction of research articles. However, following your comment, we inserted in Introduction the objectives of our work (lines 40-52).

Comment 3 page 3 (lines 91-92)

A more detail explanation was written in lines 110-114.

Round 2

Reviewer 2 Report

N/A